# Chiroptical Sensing: A Conceptual Introduction

**DOI:** 10.3390/s20040974

**Published:** 2020-02-12

**Authors:** Ani Ozcelik, Raquel Pereira-Cameselle, Natasa Poklar Ulrih, Ana G. Petrovic, José Lorenzo Alonso-Gómez

**Affiliations:** 1Department of Organic Chemistry, University of Vigo, 36310 Vigo, Spain; ozcelik@uvigo.es (A.O.); raquel@uvigo.es (R.P.-C.); 2Department of Food Science and Technology, Biotechnical Faculty, University of Ljubljana, Kongresni trg 12, 1000 Ljubljana, Slovenia; natasa.poklar@bf.uni-lj.si; 3Department of Biological & Chemical Sciences, New York Institute of Technology, New York, NY 10023, USA

**Keywords:** chiroptical systems, theoretical simulations, chiral design, sensing applications

## Abstract

Chiroptical responses have been an essential tool over the last decades for chemical structural elucidation due to their exceptional sensitivity to geometry and intermolecular interactions. In recent times, there has been an increasing interest in the search for more efficient sensing by the rational design of tailored chiroptical systems. In this review article, advances made in chiroptical systems towards their implementation in sensing applications are summarized. Strategies to generate chiroptical responses are illustrated. Theoretical approaches to assist in the design of these systems are discussed. The development of efficient chiroptical reporters in different states of matter, essential for the implementation in sensing devises, is reviewed. In the last part, remarkable examples of chiroptical sensing applications are highlighted.

## 1. Introduction

The concepts related to mirror-image symmetry have become a prominent topic as scientists have made progress in the structural determination of three-dimensional (3D) objects from both atomic and molecular levels. Particularly, the presence or lack of mirror-image symmetry plays a crucial role in shedding light on the origin of biological processes relevant to human life [1]. At different scales, objects are said to be chiral when they do not coincide with their mirror image. Accordingly, achiral objects are those possessing at least one improper axis, i.e., a plane of symmetry or a center of inversion (Figure 1) [2].

It was not until 1884 that the discovery of molecular chirality was achieved by Louis Pasteur when he observed left- and right-handed crystals of sodium ammonium tartrate tetrahydride [3]. Today, it is known that a pair of opposite-handed-shapes are called enantiomers and their interaction with other chiral entities may give rise to distinct physiological and/or toxicological responses. As an example, thalidomide was used for the treatment of morning sickness in pregnancy in the late 1950s and early 1960s. While (*R*)-thalidomide was an effective analgesic, its enantiomer (*S*)-thalidomide had a teratogenic effect, causing over 10,000 birth defects (Figure 2) [4].

Like molecules, light can also be chiral. In fact, circularly polarized light can be right-handed or left-handed, referred to as right circularly polarized light (R-CPL) and left CPL (L-CPL), respectively. These chiral waves can be generated by the summation of two perpendicular, linearly polarized lights with the same amplitude and frequency, yet, exhibiting π/2 phase difference. Moreover, a linearly polarized light emerges from the combination of two enantiomeric CPLs.

Chiroptical spectroscopies, which emerge from the difference in absorption and speed of R-CPL and L-CPL upon interaction with a chiral molecule are phenomena known as birefringence and circular dichroism (CD), respectively. The outcome of birefringence is the rotation of the plane polarized wave when traveling through a chiral system, commonly known as optical rotation. While originally these measurements were performed only at 589 nm, the D-line of sodium, resulting in the $\alpha_{D}$ values characteristic of any chiral molecule, later on, the measurement of the optical rotation at different wavelengths provided optical rotatory dispersion (ORD), a more reliable methodology for structural elucidation [5,6]. On the other hand, CD can be measured in the infrared region, as is the case for vibrational CD (VCD), or in the ultra violet visible region of the spectrum, with spectroscopies such as electronic CD (ECD), circularly polarized luminescence, or Raman optical activity [7]. In general, the chiroptical responses of enantiomers are of equal magnitude, but of the opposite sign to each other [8].

The generation of chiroptical responses is very general, ranging from monodisperse to polydisperse systems, individual or collective chirality, of organic, organometallic, or inorganic nature, or even a combination of them. While the absolute configuration determination has been the most frequently applied after the discovery of chiroptical responses, the general applicability of such methods is wide and now also includes the conformational assignment of the responding systems, detection of chiral or achiral molecules, as well as the characterization of self-assemblies, as will be described below (Figure 3) [9]. Often, the use of more than one chiroptical response is required in order to confidentially determine the structure of a particular system [10]. Furthermore, the wavelength of the induced chiroptical sensor readout is typically free of the interfering signals coming from the sample under investigation. This spectral extension is advantageous in medical applications since long exposure to light radiation can affect bimolecular samples.

The intent of this review is to provide the reader with a general overview of the key aspects to be taken into consideration when developing a chiroptical system for sensing. It starts with an illustrative description of the most commonly used strategies for the generation of chiroptical responses. Since the signatures of the chiroptical responses are not trivial, different approaches for their prediction have been developed, and here we summarize the most relevant ones. A desired application may require of chiroptical systems to be in a particular state of matter, and therefore some examples herein were classified concerning this aspect. Finally, some demonstrative examples of chiroptical sensing are outlined as related to molecular detection, switches, and diagnosis.

## 2. Strategies for the Generation of Chiroptical Responses

The rise of chiroptical methods, the key spectroscopic means of probing and reporting the presence of chiral motive(s) within the molecular framework, came as a result of coinciding advances in instrumentation as well as computer technology and algorithms that allow implementation of ab initio theoretical simulations. In a timely manner, the scientific community has made notable advances in the intelligent design of chiral systems capable of manifesting chirality in different ways: by selecting a chromophore providing chiral signals in a specific spectral region or by fine-tuning the chiroptical response via methodical selection of the preferred symmetry/geometry of the system in a given media.

Historically, the synthesis and isolation of chiral molecules has routinely incorporated measurements of OR at the single sodium-D-line wavelength (589 nm) and ECD spectral signature. To achieve a reliable stereochemical structural elucidation, nowadays, a simultaneous use of more than one chiroptical method is strongly encouraged: ORD, ECD, VCD, and ROA [11,12]. 

Different chiroptical methods probe chiral systems under variable conditions and with different forms of energy, hence accessing unique sensitivities to different structural features. Specifically, ultraviolet-visible (UV-Vis) linearly polarized light is used in the case of ORD, while circularly polarized light in the case of ECD. VCD extends the functionality of ECD to the mid-infrared regime where normal mode vibrational transitions are observed. Like VCD, ROA is a form of vibrational optical activity that is sensitive to chirality associated with all fundamental vibrational normal modes. ROA measures a small difference in the intensity of vibrational Raman scattering from chiral molecules exposed to R-CPL and L-CPL incident light. It is considered as insightful probe of the structure and behavior of biomolecules in aqueous solution [13,14]. As such, different methods require different solvation environments, different concentrations, different consideration of solute-solute vs. solute-solvent interactions, as well as intra vs. intermolecular interactions. 

In general, merits and limitations of a given chiroptical method can be identified when multiple methods are applied simultaneously to cross-examine a given stereochemical structural objective. As mentioned earlier, while OR at 589 nm is routinely reported, for structural elucidation it is advisable to consider the overall ORD pattern that extends over multiple wavelengths. While ECD requires a UV-Vis chromophore in order to produce a chiroptical response, in the case of VCD each 3N-6 normal mode vibration has potential to serve as a chirality probe of N-atomic molecule. VCD typically accesses multiple well-defined bands, yet it is less sensitive than ECD as reflected in the need for higher concentrations and 3–5 orders of magnitude due to the lower response intensity [15]. Chiroptical responses depend on the spatial orientation of chromophoric groups, moieties near the stereogenic centers, and consequently on the overall molecular flexibility. Nonetheless, VCD is highly conformation sensitive, which is why, in the case of flexible molecules, it can be occasionally hindered by conformational averaging that leads to the attenuation of the bands [14]. To reiterate, the best practice is to resort to a multiple chiroptical method in order to increase the confidence level of the assignment [11].

Since chirality can be manifested in many different ways, in this section we group chiroptical systems originating form three different strategies: individual, which is exhibited by helical molecules [16], or polymers [17], or helical nanoparticles [18]; collective, where chirality is originating from the relative orientation of individual molecules [19] or nanoparticles [20]; and induced chirality, were a chiral or achiral molecule undergoes a conformational change to adopt a chiral conformation (Figure 4) [21]. Other approaches to render chiroptical systems such as integrated photonics [22], are not in the scope of this review.

### 2.1. Individual Chirality

In this section, we want to call the attention to systems that independently present chirality. Considering individually chiral systems one may identify them as being from molecular or nanoparticle nature (Figure 5). Very often, the open molecular systems are monodisperse, like enantiopure alleno-acetylenic oligomers [10], however there are several examples of open polydisperse oligomers presenting chiroptical responses, for instance poly(phenylacetylene) amines [23]. The search for specific molecular recognition sites and restriction of the conformational space has resulted in the development of cyclic, like spirobifluorene macrocycles [24], cage-like organic, such as covalent organic helical cages [21], and organometallic systems, like alleno-acetylenic helicages [25].

On the other hand, a tremendous expansion in the last years has been realized in the development of intrinsically chiral metal nanoparticles. While for clusters, the system can be monodisperse like Au_25_ [26], larger nanoparticles are typically polydisperse and can be classified as 2D chiral nanoparticles where the chirality comes from the nanoparticle lying onto a surface, or 3D chiral nanoparticles. For a more specific review on chiroptical nanoparticles we refer the reader to reference [27].

### 2.2. Collective Chirality

Collective chirality is referred to the global asymmetry featured by 3D complex nanoarchitectures. The asymmetric organization of chiral or achiral entities mainly stems from specific weak interactions through space rather than covalent bonding. This structural arrangement can be realized by the self-assembly of independent molecules. The chirality is transfer from the individual units to the self-assembly or by template strategies where achiral units adopt relative chiral orientations by following the guidance of a chiral support (Figure 6).

To reach collective behavior in a variety of complex nanostructures with chiral morphology, bottom-up approach has been utilized for many decades. As an example, the gel formation of a chiral azobenzene via π-π stacking was attributed to be mainly responsible for the observed chiroptical responses since the isolated molecular system employed is practically CD silent [28]. On the other hand, template-guided self-assembly involves the interaction between entities and templates to tackle challenges in precise control of self-assembly at nanometer scale. For instance, well-defined arrangement of plasmonic nanoparticles can be achieved when inorganic silica helices are used as a chiral template. This complex architecture features remarkable chiroptical responses as a result of the helical arrangement of the metal nanoparticles [29].

### 2.3. Induced Chirality

The expression of chirality in achiral molecules and molecular assemblies is known as induced chirality. Typically, chiral induction is achieved when units comprising the system are arranged in an asymmetric or helical manner, even though some of the constituents of the system are achiral. From mechanistic perspective, chiral induction at the supramolecular level can be generated via host–guest complexation or by the formation of vividly termed sergeant-and-soldiers (Figure 7). Chiral induction and tunable helicity control in oligomers, host–guest complexes, and supramolecular assemblies is a desirable goal in view of the appealing applications in material science. Below, we present few selected systems that demonstrate the induction and amplification of supramolecular chirality from achiral molecules.

The folding of biopolymers into compact architectures capable of encapsulating ligands has inspired chemists to design artificial receptors based on helically folded oligomers possessing a hollow cavity. In many cases, oligomeric-foldamers have demonstrated a propensity to strongly express homo-chirality by providing confined environments suited to recognizing chiral guests with high enantioselectivity. The binding of chiral guests can lead to dynamically responsive helical chirality of an oligomeric-foldamer which serves as the host [30,31]. As an example, Moore and coworkers [32] reported that addition of chiral monoterpenes such as (–)-*R*-pinene (Figure 8) results in induction, increase, and eventual saturation of the CD signal for oligomeric phenylene ethynylene (Figure 9). While in the absence of pinene, oligomer exhibits no CD, an incremental addition of enantiomerically pure ɑ-pinene induces a strong Cotton effect in the wavelength range where the oligomer absorbs. The mirror image CD spectrum depending on addition of (–)-*S*-pinene vs. (+)-*R*-pinene, demonstrates that the induced CD signals are originating from an association of the oligomer with the chiral ɑ-pinene rather than some impurity. It is worth noting that the binding of ɑ-pinene to dodecamer was determined to be a solvophobically driven process, optimized at 40% water in acetonitrile. The uncomplexed foldamer is presumed to exists as a dynamic racemate of enantiomeric *M* and *P* helical conformations. Upon complexation to a chiral monoterpene, helical-sense bias is produced. According to authors, the induction of a strong CD signal, the magnitude of binding association constant 6830 M^‒1^, and 1:1 stoichiometry all suggests that complexation between the monoterpene and the oligomer is occurring in a specific manner and is not simply the result of nonspecific associations. Molecular models reveal that the size and shape of *R*-pinene are complementary to the internal space of the hydrophobic cavity of the foldamer. It was established that the binding strength can be attenuated by modification of the helix cavity by filling it with, for instance, methyl groups which reduce the space available for guest binding. These observations support the idea that the complex involves binding of the chiral guest within the tubular cavity, rather than along the oligomer backbone or via noncovalent domino effect at the end of the backbone.

Further developments in designing foldamers susceptible to helical-bias have included higher degree of modularity of such receptors [33]. As an illustrative example, Huc and co-workers [33] demonstrated that hollow aromatic oligoamide foldamers can form single molecular capsules when the helix diameter at each end of the sequences was decreased to close the cavity. Specifically, each monomer within the oligomeric sequence was chosen to achieve a predictable change in binding properties, thus fine-tuning the host-guest interactions. When chiral guests such as D- or L-tartaric acid were introduced to aromatic oligoamide, the achiral helical host adopted a unique handedness as evidenced by the CD signature (Figure 10). In such helical capsules, the binding and release of guests to and from the cavity require a local unfolding of the helix, possibly induced by changes in solvent, temperature, or pH.

Another type of induced chirality within host–guest complexes occurs when chirality is transferred from chiral non-chomophoric systems to chromophoric groups that adopt chirality and serve as chiral sensors. A class of achiral molecules susceptible to such chiral imprinting are metalated bis-porphyrin tweezers. A wide array of dimeric porphyrin hosts has been developed and applied towards the absolute configuration (AC) assignment of several classes of mostly bifunctional chiral compounds such as diamines, amino alcohols, or amino acids [34,35].

Upon formation of 1:1 host–guest complex through bidentate metal coordination, chirality is transferred from the guest to the host with the effect of inducing a preferential chiral twist in the porphyrin–porphyrin arrangement (Figure 11a) [36,37]. The sign of the Soret CD couplet is related to, and diagnostic of the AC of the guest (Figure 11b). Also, chiral induction via intermolecular H bonding of porphyrin oligomers rendered remarkable chiroptical responses [38].

On the other hand, diverse achiral guests such as nonchromophoric [25] or organometallic sandwich [39] can be detected due to chiral induction through inclusion complex formation with helical molecular cages.

The sergeant-and-soldier induction mechanism involves a small amount of chiral material (sergeant) that enforces a chiral structure on assembly composed predominantly of achiral molecules (soldiers) as dictated by the chirality of the sergeant. To date, numerous systems exhibiting induced chirality have been reported using this principle [40,41]. One example of sergeant-and-soldier phenomena is the induced helicity of isotactic-rich poly(2-vinylpyridine) by means of its hydrogen-bonding with chiral hexahydromandelic acid. Another possibility is the decoration of the polymer backbone with chiral moieties (Figure 12) [42].

The mirror imaging of the CD spectra suggests the formation of right- and left-handed helical chains of the achiral polymer through complexation with the chiral guest [43]. Poly(phenylacetylene)s (PPAs) have also shown to present distinct chiroptical responses associated with their helicity. These systems can remarkably switch conformation upon complexation with different guests. Riguera and coworkers [44] have demonstrated that PPA copolymers containing (*R*)- or (*S*)-methoxy-phenylacetic acid (MPA) as minor chiral pendant can selectively adopt the right- or left-handed helix, in the presence of small amounts of cations, such as Na^+^, as well as donor cosolvent (Figure 13). The helical sense depends exclusively on the chiral monomer/donor cosolvent ratio, and this allows a flexible and controlled on/off tuning of the helicity of the copolymer. 

## 3. Approaches for the Prediction of Chiroptical Responses

The prediction of the chiroptical responses is not trivial, and therefore different methodologies have been developed for their prediction. Ab initio calculations have been extensively used for small and medium sized molecular systems, not far above 1000 heavy atoms. The specific level of theory may be chosen not only considering the chiroptical response under evaluation but also the size and nature of the chiroptical system. A much more intuitive method for the prediction of chiroptical responses is the exciton chirality (EC) method, this method is reliable when the chromophores mainly responsible for the response in the system are independent, no direct conjugation between them, and may interact to each other through space (Figure 14). While EC has been extensively used for interpretation of ECD signals [45], more recently it has also been implemented in the analysis of VCD spectral profiles [46]. Nonetheless, an extension of the EC concepts to VCD has to be applied and interpreted with great care to prevent possible erroneous predictions [47].

The prediction of the chiroptical responses via the EC method has been typically applied by the summation of the pairwise interactions among all chromophores. Specifically, for systems with high symmetry, a chiroptical symmetry analysis may help in the design of chiroptical systems. On the other hand, for systems presenting surface plasmon resonance, methodologies based on classical electromagnetism are often used. We refer the reader to more specific examples on the topic, as provided in reference [48].

### 3.1. Ab Initio

The exponential growth of computational power has been the key driving force that revolutionized the way chemists approach the stereochemical structural elucidation without the need for any additional chiral reagents, chemical derivatization or reference system [49]. As such, simulations related to identifying stable geometries and predicting corresponding chiroptical responses from first principles (ab initio) have become an indispensable tool for investigating chiral molecular systems. A large number of research studies have demonstrated that molecular chiroptical properties can be computed reliably, starting from first principles quantum theory, particularly density functional theory (DFT) (Figure 15).

A number of recent review articles, for instance references [13,50,51,52,53,54], document the progress made in the computational chiroptics. Simulations are used to predict and visualize stable conformations, to assign absolute configurations, to simulate and analyze chiroptical data, as well as to provide a basis for understanding their origin. Additionally, simulations can provide supporting evidence and insight into the origin of chiral induction phenomena. On the other side of the coin, since molecular modeling always yields data, one needs to be mindful not to treat molecular modeling as a black-box tool or take the quality of simulated data for granted [13,15,54]. In this section, we discuss key steps with a few examples aimed at highlighting the most critical aspects of the ab initio protocol towards chiroptical structural elucidation. 

As routine to computational chemistry, ab initio based chiroptical predictions begin with building virtual model(s) and for each model conducting conformational survey aiming to identify prevailing molecular geometries. In this initial step, empirical data from NMR spectroscopy and X-ray crystallography could yield the relative configuration (RC) of selected stereogenic centers which can significantly reduce the number of necessary virtual models. An illustrative example of valuable a priori RC determination is the study of marine natural product (+)-bistramide C, which is endowed with 10 stereogenic centers [55]. Specifically, NMR-based study has reduced the candidate pool of 1024 possible stereoisomers, to only 16 diastereomers needed for ECD theoretical consideration. In the absence of RC data, for a system with n elements of chirality, one must consider all 2n ‒ 1 diastereomers, mindful that enantomeric forms provide mirror image chiroptical responses.

The most common approach to identify the local minima is to initiate the survey of potential energy surface (PES) by resorting to molecular mechanics (MM) based molecular dynamics (MD) or Monte Carlo (MC) algorithms with a well mindful selection of parametrized force field based on atom-types and moieties (e.g., MMFFs, OPLS3, AMBER, etc.). It is of critical importance that conformational survey is conducted in a comprehensive manner in order to avoid any carryover errors from this initial step that have negative consequences on the final chiroptical results and are challenging to track-back. Even if the conformer is not highly populated, it could intrinsically provide a strong chiroptical response and, as such, have influence on the overall predicted spectral signature. Therefore, the use of multiple algorithms and possibly applicable force fields are advisable [13,53,54] in order to properly explore variation in all rotatable bonds and overcome energy barriers associated with ring-puckering modes.

The stable MM-identified geometries within ~10–20 kcal/mol energy window [56], depending on the intrinsic degrees of freedom, are subsequently fully optimized under appropriate quantum mechanical levels of theory. For larger molecular systems, it is not uncommon to first apply semi-empirical AM1 or PM3 methods, followed by full geometry optimization at a higher ab initio level of theory. DFT, which encompasses a large variety of different levels of approximation and theory, has been benchmarked as a standard computational method, yielding the best average performance for energy minimization and chiroptical predictions. Contrary to colloquial statement “one size fits all”, in choosing the appropriate DFT-based level of theory, there is no classification of molecular system by size or atom type that allows for single universal reliable approach. Every simulations method is prone to error [57]. Therefore, one should resort to exploring different combinations of DFT functionals and basis sets in the search of consistencies in Boltzmann populations of stable conformations as well as, at a later stage, corresponding chiroptical responses. Simulation results that are more consistent among each other and, most importantly, that more closely correlate with the experimental data are considered as results delivering a higher confidence level. Identification of true PES minima resulting from the full geometry optimization is verified by the absence of negative vibrational frequencies at the same level of theory.

Two suggested families of functionals that can be explored are hybrid and range-separated ones. Popular hybrid functional with increasing amounts of “exact” HF exchange integral are B3LYP (20% HF), PBE0 (25%), M06 (27%), BH&HLYP (50%), and M06-2X (54%). Range-separated functionals such as Coulomb-attenuated CAM-B3LYP and ωB97X functionals often outperform the hybrid ones in ECD predictions and, hence, are considered standard. One instructive illustrative example can be found in references [54,58]. It is important to emphasize that relative Gibbs free energies, dipole, and rotational strength can be very sensitive to the choice of functional. The literature provides several cautionary tales of studies which demonstrate that popularly used B3LYP is not necessarily the most accurate functional for predicting structures and energies of standard organic molecules, especially when noncovalent interactions come into play [13,54,58,59,60,61].

While 6-31G* can be used as the most fundamental basis set to obtain initial simulation-based insight, double-zeta-ζ or triple- ζ as well as augmented basis sets (aug-TZVP, aug-cc-pVDZ) as well as those with polarization functions (6-311++G(2d,2p)) are recommended as capable of producing quality chiroptical outcomes. If ionic species are involved, diffuse functions must be included within the basis set selection. Los Alamos National Laboratory 2 Double-Zeta (LANL2DZ) basis set [31] is extensively tested effective core potential to model metal atoms [33]. Benchmark investigation demonstrates that CAM-B3LYP with the aug-cc-pVDZ basis set can be a useful method considering both accuracy and reasonable computational time to predict the Boltzmann average optical rotation that matches the experimental, and thereby the absolute configuration of chiral molecules [62]. An illustrative case for which both geometry optimization and TDDFT calculation of chiroptical property can be highly sensitive to the used basis set is the ORD-based investigation of isocytoxazone (Figure 16) [63].

While the functional and basis set should be varied and explored for consistency, the solvent model should be accounted to match the media used for experimental chiroptical measurements. Solvent may affect the relative stability of conformers, especially with propensities for non-bonding interactions. In most cases, continuum solvent models such as conductor like screening model (COSMO) and polarizable continuum model (PCM) are satisfactory. According to few case-studies, the reliability of PCM predictions of solvent effects on optical rotations is dependent on the solvent chosen [64]. If spectral predictions are not optimal and verifying level of theory do not improve the level of correlation, one may choose to resort to explicit solvent to account for specific solute-solvent and solute-solute (dimer, trimer) interactions. For examples of such a procedure, we refer the interested reader to the literature [65,66,67,68,69].

Besides the choice of functional and basis sets, vibrational effects can play notable role in the simulated chiroptical response, especially in the case of ORD and ECD. Predicted electronic chiroptical data in which a molecule’s environment and inclusion of vibrational effects have been taken into consideration have generally improved agreement with experimental data. One illustrative example is the ORD-based structural elucidation of (R)-methyloxirane by means of a novel computational protocol, involving MD trajectory. Specifically, this method takes into account vibrational averaging and solvent effects, leading for the first time to a quantitative agreement (both sign and absolute value) between computed and experimental OR values at several frequencies [70]. This research area will remain highly active as new computational methods are being developed.

Chiroptical properties are subsequently computed from first principles and Boltzmann averaged based on predicted Gibbs free energies if more than one stable structures (ΔE ≤ 1 kcal) are identified. The variability of ab initio predicted chiroptical signals depending on conformer, underscores the importance of through conformational exploration and geometry optimization as a fundamental and one of the most important step in carrying out AC assignment [63]. In general, the weaker the experimental chiroptical response and/or the more conformationally flexible the molecular system under study, the more likely will be its dependence on the calculation method, including functional, basis set, and solvation model. As previously mentioned, TDDFT has been established to be main state-of-the-art technique to obtain accurate numerical results for chiroptical predictions. In addition, methods that have been used to provide a trustworthy chiroptical spectroscopic response are time dependent HF (TDHF) [21] and more computationally demanding coupled cluster (CC) method. For larger systems with up to 1000 atoms, the simplified TDDFT (sTDDFT) method has been recently introduced as good combination between computational cost and satisfactory accuracy for ECD prediction [71].

Typically, when ECD and VCD spectra are simulated, the computed “stick” spectra corresponding to vertical electronic and vibrational transitions are broadened with Gaussian and Lorentzian functions, respectively. ECD spectra often require a global shift of the excitations, because of the tendency of TDDFT to underestimate excitation energies [13,20]. Therefore, the excitation energies are typically shifted by +0.2 eV to +0.45 eV. On the other hand, due to the variation method nature of the approach of calculating VCD signals, vibrational energies are typically overestimated and need to be downshifted by factors recommended depending on the given basis set level of theory. 

It might appear that ab initio predictions of multiple chiroptical spectroscopic methods are likely to give redundant structural information. However, at the very least, simulations of multiple chiroptical responses serve as independent verifications of molecular structures. In general, the weaker the computed chiroptical spectrum, the stronger the dependence on the level of theory (functional, basis set, and solvation model selection). When a given chiroptical spectroscopic method gives ambiguous results, the use of more than one chiroptical spectroscopic method may provide missing information and increase the overall confidence level of the stereochemical assignment.

It is worth mentioning that, in the past decade, it has been brought to light that traditional solely qualitative correlation between experimental and ab initio predicted spectral profiles can provide limiting and even misleading analyses as such approach overlooks the valuable stereochemical information embedded within the corresponding electronic absorption (EA) as well as electronic dissymmetry factor (EDF) spectra. Therefore, one of the recent trends pioneered by Polavarapu et al. is to correlate theoretical vs. experimental ECD, EA and EDF spectra in order to enhance applicability as well as increase confidence level of stereochemical structural elucidations via electronically based chiroptical spectroscopy. As such, quantitative correlations between experimental and theoretical electronic signatures represent the suggested overall ab initio protocol. Algorithms have been implemented in general-purpose programs such as CDSpecTech which helps generate, cross-correlate, and quantitatively score the degree of overall between theoretical and experimental EA, ECD, and EDF spectra. We refer readers to further literature for recent case studies which demonstrate the scope of applicability and benefits of applying the novel quantitative ECD approach towards reliable stereochemical elucidations [72,73,74].

### 3.2. Exciton Chirality

When two equivalent chromophores are present in the same system, an a priori degenerated associated electronic transitions may split into two nondegenerated in-phase and out-of-phase transitions. Figure 10 shows the electron transition dipole moment (ETDM) associated with a chromophore and the two possible associated transitions for systems incorporating two of them. Additionally, if the two chromophores have a mutual chiral arrangement, the specific chiroptical signature can be easily associated with such chiral arrangement [75]. Particularly, if the Cotton effect of lower energy is positive and the other negative, the torsion angle between the two associated ETDMs is positive, and the opposite is true for the contrary. This simple relation between the CD response and the geometry of the chiroptical system has been extensively used not only for the relative and absolute configuration of complex systems, but also for the detection of molecular systems [76].

For systems presenting more than two independent chromophores, the chiroptical responses may be determined by the summation of the chiroptical responses originated for pairwise coupling between all chromophores. However, if the system presents higher symmetry, a chiroptical symmetry analysis can be emploied as alternative (Figure 17) [77]. For instance, for systems presenting *D*_3_ and *D*_4_ symmetry bearing three and four isolated crhomophores respectively, only one A_2_ transition and two degenerated E transitions emerge. As in the case of exciton coupling between two independent chromophores, the prediction of the chiroptical spectrum for these systems can be performed by the following steps: i) identification of the chromophores originating the chiroptical responses, ii) using group theory, obtain the symmetry-based irreduible representations to determine the allowed transitions, iii) considering the relative orientation between the chromophores, and using the Davydov´s equation, determine the energy differene between the different transitions. iv) with the geometric parametes, predict the electronic and magnetic dipole moment of each transition, and consequantly the rotary strength. As a case study, the chiroptical symmetry analysis of trianglimines has been recently performed [78].

## 4. Chiroptical Systems on Different States of Matter

There is a vast amount of chiroptical responding systems, few of which have been employed in chiroptical applications. Among the requirements the applicability of chiroptical systems, one needs to take into account the state of matter required for a specific application. 

Therefore, we want to draw the attention of the reader in this section to chiroptical samples that are in solution and on two-dimensional surfaces (Figure 18).

### 4.1. Chiroptical Solutions

Chiroptical solutions are by far the most abundant among chiroptical systems. The g-factor, the ratio between circular dichroism and absorption, is typically used as a measure of the chiroptical power. Therefore, this parameter is often used to evaluate the chiroptical responses. The highest reported for purely organic molecules is 0.05 for polyaromatic systems [79], 0.01 for alleno- [80] and spiro-acetylenic [24] oligomers, 0.06 for protein complexes [81], and 0.03 for pillar-arene systems [82,83]. On the other hand, colloidal small metal nanoparticles present typically values below 0.001 [84] while fluid suspensions of nanorods may reach a g-factor c.a. 0.022 [20].

### 4.2. Two-dimensional (2D) Chiroptical Surfaces

Chiroptical methods have been extensively used for the structural characterization of chiral systems and complexes due to their high sensitivity as mentioned above. To expand their applicability in today’s world, the development of 2D surfaces with tailored chiroptical properties is essential. Such complex systems are described as those nanostructures lacking mirror-image symmetry in solid state. Among other strategies, the inherent chirality of either small or large molecules could be transferred to achiral metal substrates upon self-assembly. Regarding the former case, Wälti and co-workers conducted a conformational study on monolayers of synthetic peptide by means of CD spectroscopy [85]. On the other hand, unlike large molecular systems, the exploration of chiroptical responses featured by single monolayers of small molecules might require more sophisticated chiroptical methods. In this regard, the chiral 2D ordering of allenes on a metal surface has been previously studied by scanning tunneling microscopy, yet the low stability hampered the exploration of the chiroptical responses of the monolayer [86]. Recently, the incorporation of anchoring groups to the allenic moiety has enabled the formation of device-compatible chiroptical surfaces (Figure 19).

Whereas the conventional CD spectroscopy measurements have yielded ambiguous results, the chiroptical responses of these molecule-thin sheets have been addressed via second harmonic generation spectroscopy on a custom-made transparent substrate [87].

## 5. Chiroptical Sensing Applications 

Chiral sensing and analysis are important to provide unique functionalities for many biomolecules, drugs, and natural products. The limitation of wider applications of chiral sensing in biological and medical applications arises from the nature of the chiral signals, which are inherently weak. In this section, we present the current applications of chiroptical sensing, which include molecular detection and disease diagnosis (Figure 20).

### 5.1. Chiroptical Molecular Detection

Many new applications of chiroptical sensing require sensitivities beyond the current limits. The new horizons of chiroptical sensing applications based on single-molecule chirality determinations are in the field of protein structure determination in situ (e.g., in solution, at surfaces, within cells and membranes) and in the field of metabolomics in medicine, for diagnostic purposes (e.g., chiral analysis of body fluids). These have been coupled with the development of new chromatographic separation techniques and standards for the chiral identification of components in complex mixtures, for analytical purposes in pharmacy and chemistry. 

Chiral sensing processes require two steps: molecular sensing and signal transduction. The tremendous progress in nanotechnologies and nanomaterials has presented new opportunities for the development of intelligent chiral sensors. For nanotechnologies, the new concepts available for designing new chiral sensors include atomic switches, probe-fabrication of molecular arrays, and integrated circuit technology. These can be coupled with the newly available nanostructures, which include nanotubes (i.e., carbon nanotubes) and nanosheets, nanoparticles, nanorods, nanowires, nanowhiskers, mesoporous silica, mesoporous carbon and other mesoporous materials, organic-inorganic nanohybrids, and bio-related nanohybrids [88]. 

In recent years, the development of chiroptical sensors has been orientated toward metamaterial and plasmonic platforms for manipulation of local fields, with the purpose of being able to enhance chiroptical signals. Metamaterials are used for chiral biomolecules with optically active bending at ultraviolet (UV), visible, and infrared (IR) wavelengths [48]. Metamaterial platforms based on their structure can be chiral or achiral. Chiral sensing platforms have an advantage due to the creation of strong optical chirality in the vicinity of chiral metamaterials, and direct detection of large chiral molecules. The first metasurface with a single layer of gammadion structure was used to detect various proteins and the amino acid tryptophan [89]. The measurements were based on the spectral displacement in the far-field spectrum due to near-field interactions of chiral molecules with the metasurface. Later on, the same mechanism was applied to IgG and different smaller chiral molecules, by applying different metamaterials [90,91,92,93].

The main drawback of chiral metamaterial platforms is the strong CD signal that is generated by the platforms in comparison with the relatively weak molecular signal. To overcome these limitations of chiral metamaterial platforms, achiral metamaterials have been used instead. The CD signal of a chiral molecule is enhanced around the resonance frequency in a plasmonic nanosphere, which results in the CD signal in the millidegree range being extended to visible wavelengths. The advantage of this spectral extension to the visible range is to avoid the exposure of biomolecular samples to UV radiation. However, achiral plasmonic structures prevent unambiguous chiral detection due to the sensitivity of the CD signals to the binding orientation of the chiral molecule regarding the plasmonic nanoparticles [94].

On the other hand, the building of nanosized supramolecular host complexes using the self-assembly of molecular components has extensive applications in many fields of technology, chemistry, material sciences, and sensor development. The nanosized supramolecular structures are stabilized by multiple, weak, and noncovalent interactions, which include hydrogen bonding, electrostatic, van der Waals forces and hydrophobic effects, π-π stacking interactions, and metal coordination. Based on the nature of the molecules and different weak interaction pathways, molecules can self-assemble into different nanostructure motifs, such as tubes, rods, and sheets [95]. Porphyrins, metalporphyrins, and their assemblies can form a large number of nanomaterials, like rods, rings, particles, sheets, wires, and tubes, which have different electronic and structural characteristics [96]. The presence of a chiral cationic functionality at the porphyrin periphery results in solid-state systems presenting elements of supramolecular chirality. For example, a tetraphenylporphyrin bearing an (L)-prolininium fragment forms a large porphyrin aggregate with high supramolecular chirality, as shown by the CD spectrum. This material shows high sensitivity to limonene [97]. In solution, achiral diporphyrins have also been used extensively for chiroptical detection of a large diversity of synthetic chiral compounds and natural chiral products [35,65]. Similarly, ß-cyclodextrin has a well-defined hydrophobic inner cavity and a hydrophilic shell, and it is an appropriate host for forming supramolecular structures. β-Cyclodextrin can form host–guest inclusion complexes through selective combinations of miscellaneous inorganic and organic molecules and biomolecules into its cavities. The inherent chirality of this system was used for the chiroptical detection of resveratrol [66]. Helical organometallic [25] and purely organic cages [21,39] have been used for the chiroptical detection of diverse molecular systems. Guest chirality could be monitored by ECD upon complexation with self-assembly based rotaxanation of cyclodextrines [98]. Circularly polarized luminescence has been used to monitor the stoichiometric-controlled inversion of chiral gelators with and achiral tetraphenylethylene [99].

### 5.2. Chiroptical Diagnosis

Plasmonic nanoparticles are materials with great potential for applications in biomedicine, as biosensors and drug delivery systems, as well as in photothermal therapies [67]. In this section, we will present a few successful cases of practical uses of chiroptical sensors in the diagnosis of diseases.

Prostate-specific antigen is a specific biomarker for prostate cancer. Tang developed a highly selective chiroptical detector of prostate-specific antigen by using gold nanorod dimers assembled via complementary DNA fragments. The CD signals were amplified using a silver shell deposition on the surface of the gold nanorod dimers. This biosensor was postulated to serve as a versatile methodology for cancer biomarkers [68].

Neurodegenerative disorders are characterized by the presence of amyloid plaques that are composed of misfolded proteins, as seen for Parkinson’s disease, Alzheimer’s disease, transmissible spongiform encephalopathies, and amylotrophic lateral sclerosis. Therefore, there is the need to develop diagnostic tools that will allow detection of the specific forms of these protein aggregates. This will improve the early treatment of such diseases, and provide further understanding of the fundamental aspects of these neurodegenerative disorders. Plasmonic golden nanorods have been successfully used in the detection of amyloid plaques in Parkinson’s disease and in prion diseases. The advantage in the use of nanorods is that they do not react with monomeric proteins, but are absorbed onto protein fibrils. Chiral amyloid templates, therefore, induce the arrangement of nanorods yielding the increase of optical activity at the plasmon resonance wavelengths [55].

Type 1 diabetes mellitus is characterized by increased concentrations of blood glucose [100]. Additionally, the levels of other biomarkers associated with the pathogenesis of diabetes are changed. The conformation of blood plasma proteins (i.e., their 3D structure) and other biomolecules are also changed. However, the observation of these structural changes by conventional spectroscopic techniques is limited, including the use of Raman and infrared spectroscopy. Chiroptical spectroscopy is sensitive to any structural changes to chiral molecules, so it can be used to sense any small structural changes. Therefore, the identification of spectral biomarkers based on biomolecular structural changes using chiroptics might be a promising complement to conventional diagnostic methods for type 1 diabetes mellitus [83]. 

To date, most of the plasmonic sensors and systems are rigid and passive. Jeong and co-workers developed dynamic plasmonic nanoparticles, which can be employed as in situ mechanical probes for rheological properties of a fluid at nanoscale with microscopic volumes. They used chiral magneto-plasmonic nanocolloids, which can be actuated by an external magnetic field, to allow direct and rapid modulation of their distinct optical responses. This is an example of noninvasive measurements using optical plasmonic sensing methods that will have tremendous applications in medicine (e.g. in-situ active nanorheology for viscosity measurements of complex biological fluids, such as blood plasma) [101].

## 6. Conclusions and Perspectives

This review provides an overview of important methods for chiral sensing via chiroptical spectroscopy and addresses possible mechanisms for chiral design that lead to the generation and amplification of the chiroptical response. Such chiral design and sensing applications are becoming increasingly important in all areas of chemistry, biochemistry, and structural biology, including implementing chiral systems of various scales and phases of mater, prominently in the arena of pharmaceutical, agricultural, and optoelectronic industries. When different sources of chirality, such as individual, collective, and/or induced chirality are present in a molecular system, it may not be possible to perform the structural analysis from an individual type of chiroptical response. Mindful of the scope of applicability, merits, and limitations of a given chiroptical method can be identified when multiple methods are applied simultaneously to cross-examine a given stereochemical structural objective. Some pharmaceutical companies have established a facility often termed as “chiral tool box” which contains chiroptical spectroscopic equipment and related computational methods for analyzing the chiroptical responses. Particularly promising for advances in medicinal fields are novel case studies, presented herein, which demonstrate chiral design and sensing as the frontier in chiroptical diagnosis. We hope that this review will provide a source of inspiration for novel types of chiral design and amplification mechanisms for current practitioners as well as serve as a valuable reference for future generation scientists.

## Figures and Tables

**Figure 1 sensors-20-00974-f001:**
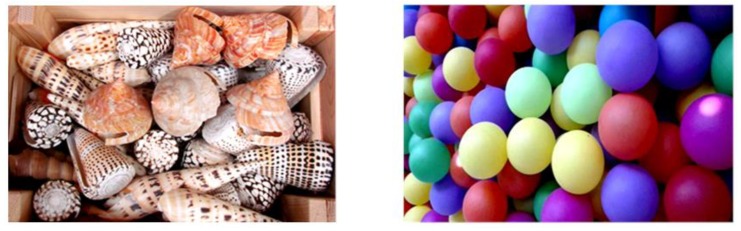
Chiral (**left**) and achiral (**right**) macroscopic objects.

**Figure 2 sensors-20-00974-f002:**
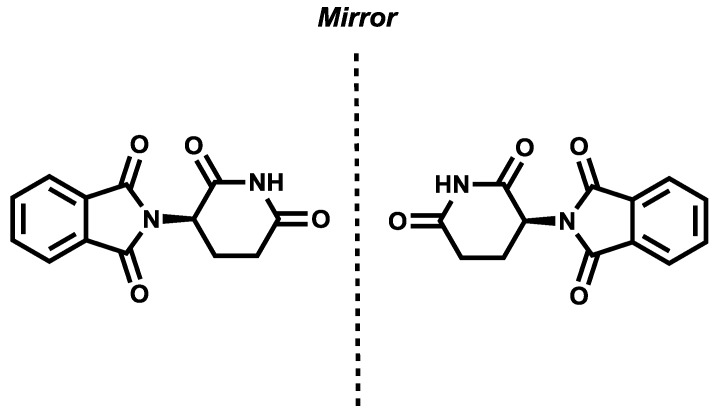
(*R*)-Thalidomide (**left**) and (*S*)-Thalidomide (**right**).

**Figure 3 sensors-20-00974-f003:**
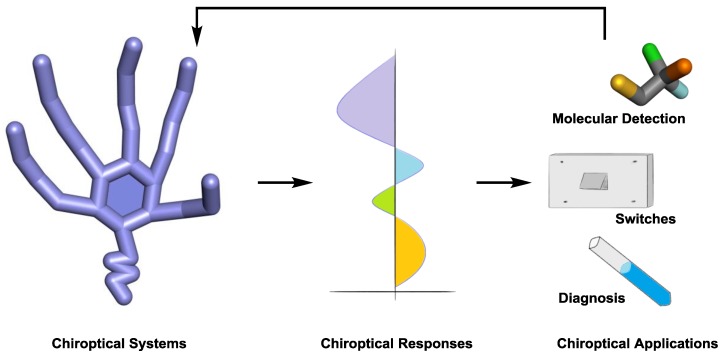
General representation of the applicability of a chiroptical system.

**Figure 4 sensors-20-00974-f004:**
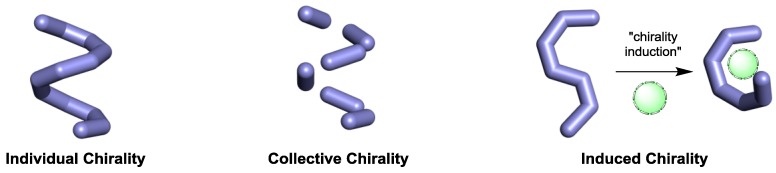
Different strategies to chiroptical systems.

**Figure 5 sensors-20-00974-f005:**
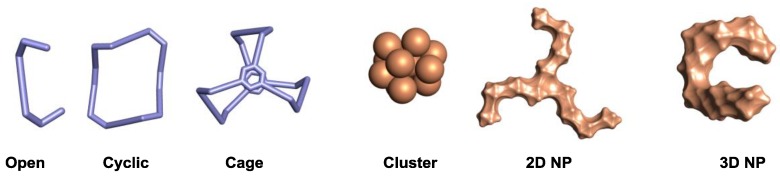
Different examples of systems presenting individual chirality.

**Figure 6 sensors-20-00974-f006:**
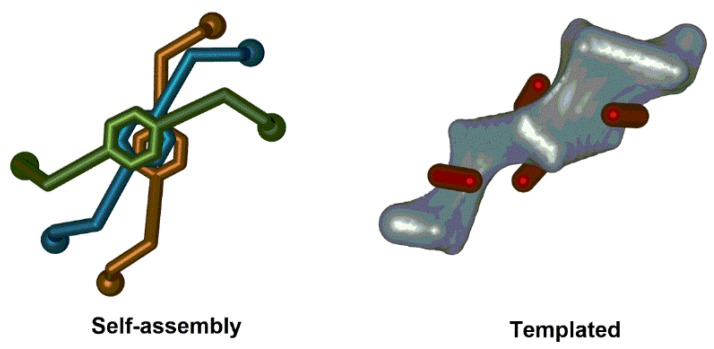
Representation of self-assembly and template-guided self-assembly processes to reach collective chirality in 3D nanoarchitectures.

**Figure 7 sensors-20-00974-f007:**
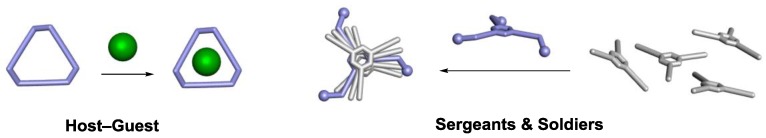
Representation of host–guest and sergeants and soldiers strategies for chiral induction.

**Figure 8 sensors-20-00974-f008:**
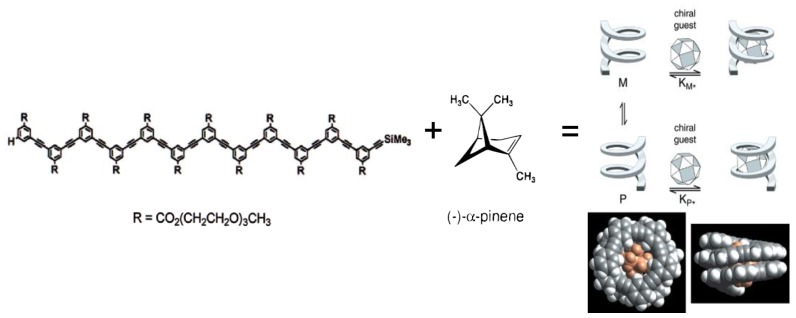
Association models of oligomeric phenylene ethynylene foldamer encapsulating (-)-R-pinene [32].

**Figure 9 sensors-20-00974-f009:**
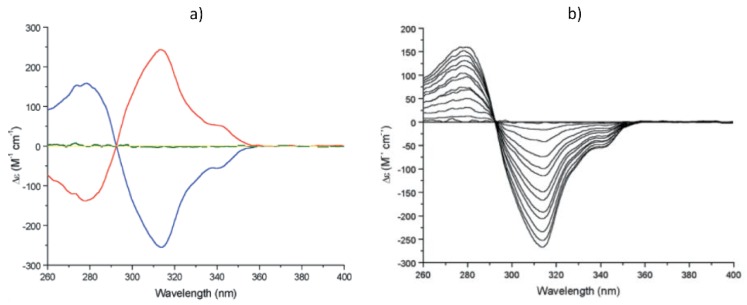
(**a**) CD spectra of (–)-R-pinene (yellow line), oligomer (green line) in the presence of 100 equiv of (–)-R-pinene (blue line) and 100 equiv of (+)-R-pinene (red line). (**b**) CD spectra of oligomer 1 as a function of the (–)-R-pinene concentration [32].

**Figure 10 sensors-20-00974-f010:**
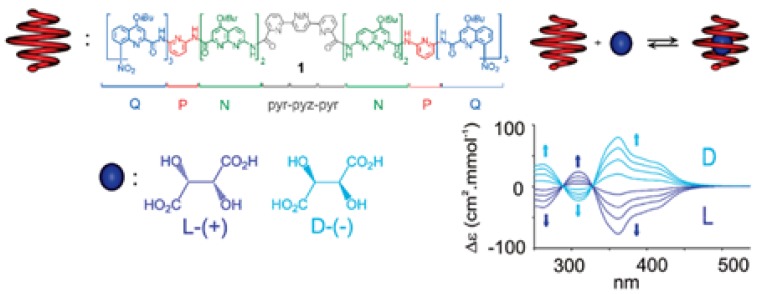
CD evidence of diastereoselective encapsulation of tartaric acid by a helical aromatic oligoamide [33].

**Figure 11 sensors-20-00974-f011:**
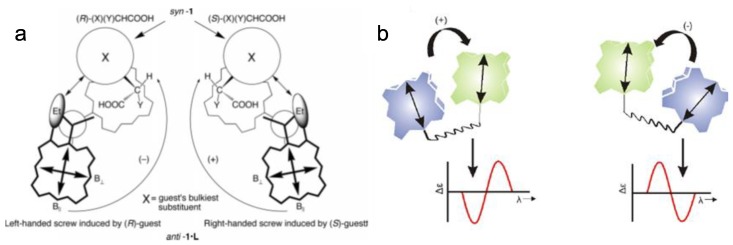
(**a**) Supramolecular chirogenesis in a bis-porphyrin tweezer upon interaction with a chiral acid. (**b**) Exciton coupling of porphyrins with relative positive or negative twist [35,36,37].

**Figure 12 sensors-20-00974-f012:**
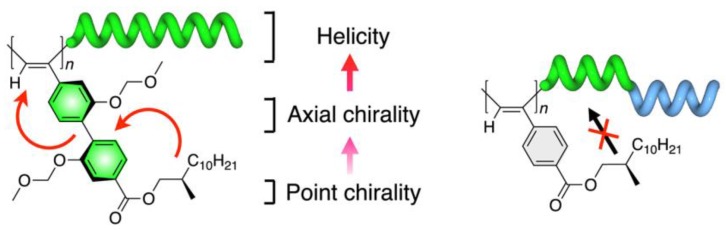
Illustration of the chiral induction form the peripherical chiral moiety to the polymer backbone [42].

**Figure 13 sensors-20-00974-f013:**
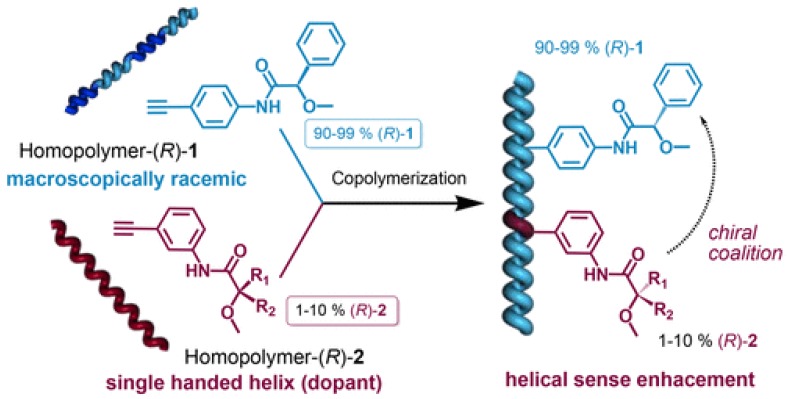
Chiral induction of PPA copolymers containing MPA, in the presence of small amounts of Na+ and MeOH as a donor co-solvent [44].

**Figure 14 sensors-20-00974-f014:**
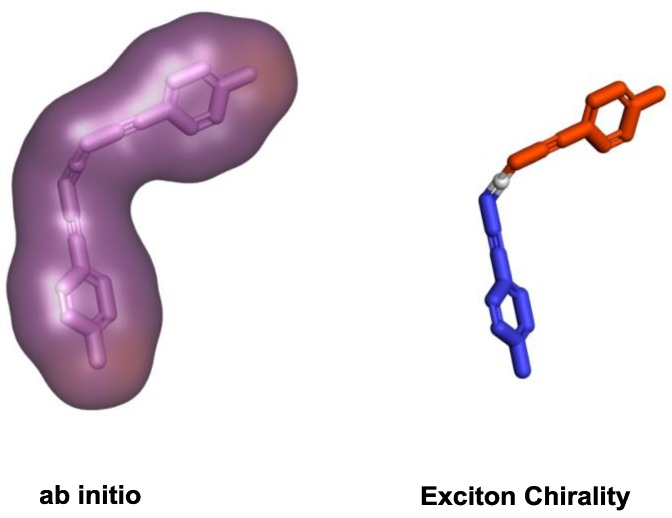
Different approaches for chiroptical prediction.

**Figure 15 sensors-20-00974-f015:**
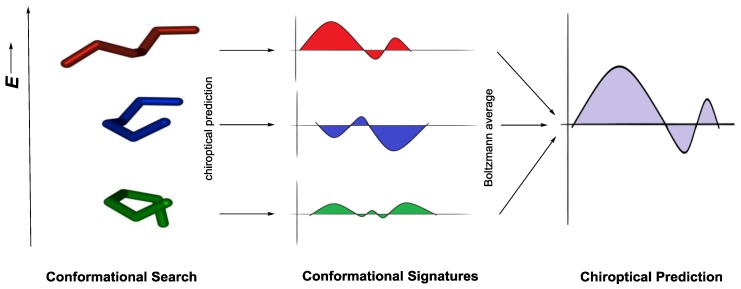
Necessary steps for the chiroptical prediction.

**Figure 16 sensors-20-00974-f016:**
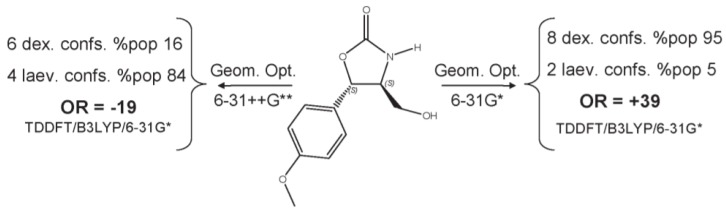
Dependence of predicted sign and magnitude of ORD for isocytoxazone based on on conformational distribution and TDDFT basis set level of theory [63].

**Figure 17 sensors-20-00974-f017:**
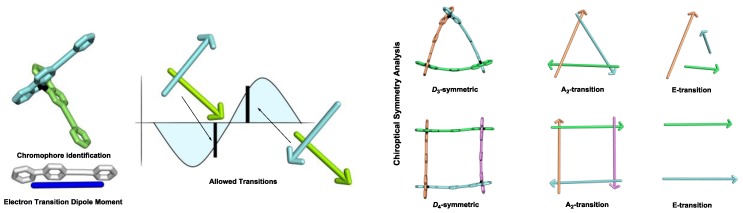
Example of exiton chirality (**left**) an chiroptical symmetry analysis (**right**).

**Figure 18 sensors-20-00974-f018:**
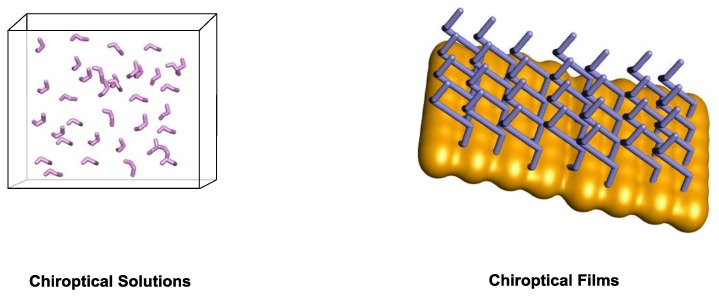
Illustration of chiroptical systems in different states of matter.

**Figure 19 sensors-20-00974-f019:**
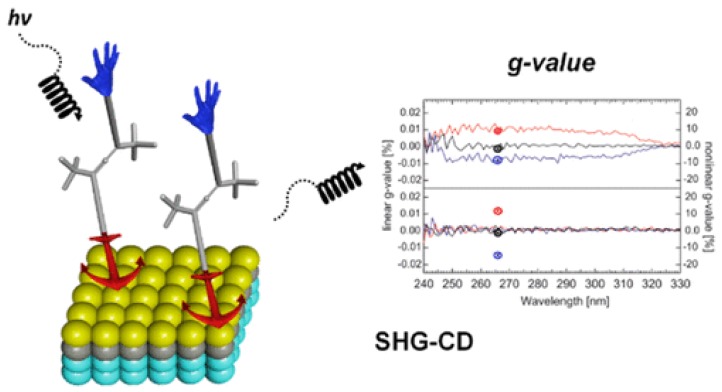
Device-Compatible Chiroptical Surfaces through Self-Assembly of Enantiopure Allenes [87].

**Figure 20 sensors-20-00974-f020:**
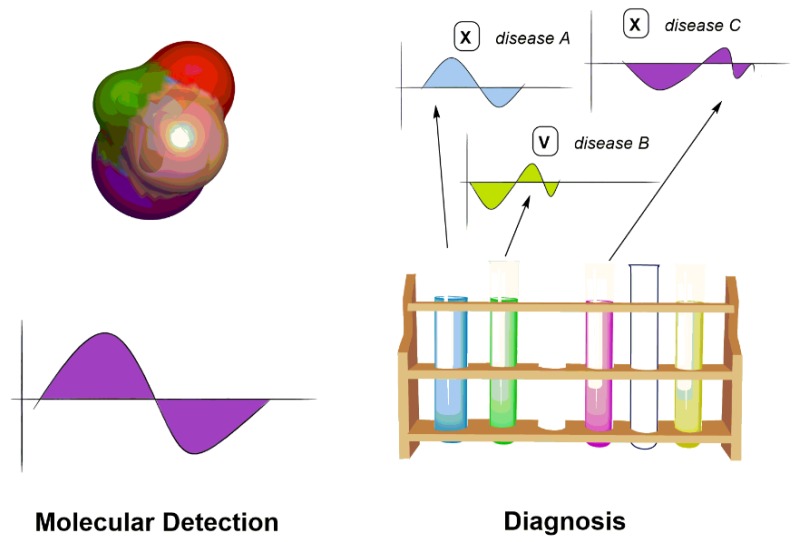
Representation of chiroptical applications.

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
