# Peer review of "Chiroptical Sensing: A Conceptual Introduction"

_sensors, 2020, doi:10.3390/s20040974_

Round 1

Reviewer 1 Report

Even though this review cites a vast literature, it is not well-written and readable for the wide community that reads this journal. Figures are of bad quality, do not contain scientific information and are all based on sketches. Even with these unattractive sketches, that feel empty and not explained enough in the text and in the questions, it is an overall feeling as if the scientific language did not existed. A reader would like to see some of the results of the cited work, presented and explained with proper units and figures of merit. Just to mention one: e.g. in Figure 9 nothing is clear, energy values, x axis, colors. This must be improved. Overall, I think that the manuscript should be rewritten and resubmitted.

Author Response

We would like to express our gratitude for the constructive input from all three reviewer 1.

“Even though this review cites a vast literature, it is not well-written and readable for the wide community that reads this journal.”

We have improved the manuscript in order to make it readable for the community of Sensors. The modifications are highlighted in the manuscript.

“Figures are of bad quality, do not contain scientific information and are all based on sketches. Even with these unattractive sketches, that feel empty and not explained enough in the text and in the questions, it is an overall feeling as if the scientific language did not existed. A reader would like to see some of the results of the cited work, presented and explained with proper units and figures of merit. Just to mention one: e.g. in Figure 9 nothing is clear, energy values, x axis, colors. This must be improved. Overall, I think that the manuscript should be rewritten and resubmitted.”

We have revised the quality of all figures. We have followed reviewer’s suggestion to integrate multiple illustrative examples from the cited work. New Figures and corresponding text provide additional scientific information that enhances the manuscript by exemplifying the overarching principles associated with chiroptical sensing. Figure 9 has been improved to help the reader. The modifications are highlighted in the manuscript.

Reviewer 2 Report

The present paper review the recent advance in the field of chiroptical sensing. This paper is more of a conceptual introduction to the field. At the same time, the authors also introduced the application of chiral sensing in different fields. The paper does not give too much description of the detailed chiroptical sensing methods. From this point of view, the summary of this field is not very comprehensive. I suggest that the author narrow down the topic of the paper. “Chiroptical Sensing” is too wide to be completely included by this short review. I suggest that the author refer to the following papers.

P12, 380-386, g factor of pillar[5]arene derivatives could be larger than 0.015 (Angew Chem Int Ed, 2017, 6869, Chem. J. Eur., 2019, 12526).

Recent chiroptical sensing studies based on molecular aggregation and recoginition should be included (Chem. Commun. 2018, 2643; ibid 2018, 9206; ibid, 2019, 3156; ibid, 2020, 161)

Author Response

We would like to express our gratitude for the constructive input from reviewer 2.

“The present paper review the recent advance in the field of chiroptical sensing. This paper is more of a conceptual introduction to the field. At the same time, the authors also introduced the application of chiral sensing in different fields. The paper does not give too much description of the detailed chiroptical sensing methods. From this point of view, the summary of this field is not very comprehensive. I suggest that the author narrow down the topic of the paper. “Chiroptical Sensing” is too wide to be completely included by this short review.”

We have changed the title of the review to Chiroptical Sensing: a Conceptual Introduction. In order to give a more comprehensive description of the chiroptical sensing methods, we have integrated specific examples. The modifications are highlighted in the manuscript.

“I suggest that the author refer to the following papers.

P12, 380-386, g factor of pillar[5]arene derivatives could be larger than 0.015 (Angew Chem Int Ed, 2017, 6869, Chem. J. Eur., 2019, 12526).

The references have been added to the manuscript as well as the following text 03 for pillar-arene systems.[68,69]. The modifications are highlighted in the manuscript.

Recent chiroptical sensing studies based on molecular aggregation and recognition should be included (Chem. Commun. 2018, 2643; ibid 2018, 9206; ibid, 2019, 3156; ibid, 2020, 161)”

We have introduced the references and included the following explanation in the manuscript: Guest chirality could be monitored by ECD upon complexation with self-assembly based rotaxanation of cyclodextrines.[84] Circularly polarized luminescence has been used to monitor the stoichiometric-controlled inversion of chiral gelators with and achiral tetraphenylethylene.[85] The modifications are highlighted in the manuscript.

Reviewer 3 Report

This review paper devoted to the field of Chiroptical Sensing. While this area is a very important branch of modern chemistry, it is rather wide and interdisciplinary ranging from theoretical and computational methods to practical applications in medicine, environmental control, etc. To date, there are several comprehensive and detailed reviews published on this subject and covered almost all main aspects of this phenomenon. Indeed, the authors also tried to include many topics in their review. However, the presented review is too concise for this task resulting in somewhat perfunctory and superficial style of the manuscript. For example, many important contributions in this field such as porphyrin-based systems by Inoue, polymer-based system by Yashima, and MCD application for unambiguous rationalization of chiroptical sensing by Kobayashi have been simply ignored. In my opinion, this review should be considerably expanded to make a logically constructed paper to include the main advantages in the field or specifically orientated to just one or two particular topics of chirality sensing to discuss these topics comprehensively.

Besides, there are some specific comments, which should be properly addressed, as follows.

There are many abbreviations, which have to be defined at the place where they introduced for the first time. Some figures are not so clear and should be re-drawn for clarity, such as "Induced Chirality" in Figure 4, "Cyclic" in Figure 5, "Host-Guest" in Figure 7, "ab initio" and "Exciton Chirality" in Figure 8, "Chiroptical Solutions" in Figure 11. Also I cannot understand the meaning to include “Classical Electromagnetism” as a separate sub-chapter and show it in Figure 8 if it is not a subject of this review. The same problem with “chiroptical symmetry analysis.” There is a large number of typos and errors in the text, which should be carefully checked.

Author Response

We would like to express our gratitude for the constructive input from reviewer 3.

This review paper devoted to the field of Chiroptical Sensing. While this area is a very important branch of modern chemistry, it is rather wide and interdisciplinary ranging from theoretical and computational methods to practical applications in medicine, environmental control, etc. To date, there are several comprehensive and detailed reviews published on this subject and covered almost all main aspects of this phenomenon. Indeed, the authors also tried to include many topics in their review. However, the presented review is too concise for this task resulting in somewhat perfunctory and superficial style of the manuscript. For example, many important contributions in this field such as porphyrin-based systems by Inoue, polymer-based system by Yashima, and MCD application for unambiguous rationalization of chiroptical sensing by Kobayashi have been simply ignored. In my opinion, this review should be considerably expanded to make a logically constructed paper to include the main advantages in the field or specifically orientated to just one or two particular topics of chirality sensing to discuss these topics comprehensively.

In order to provide a more comprehensive account for more recent advances in chiroptical sensing field, we have followed reviewer’s suggestion and incorporated examples from recent publications such as porphyrin-based systems (references Org. Lett., 9, 2007, Org. Lett., 10, 2008, and J. Am. Chem. Soc. 2005, 127, 534-535) by Inoue and polymer-based system (reference J. Am. Chem. Soc. 2019, 141, 7605–7614 by Yashima. The modifications are highlighted in the manuscript.

Besides, there are some specific comments, which should be properly addressed, as follows.

There are many abbreviations, which have to be defined at the place where they introduced for the first time. Some figures are not so clear and should be re-drawn for clarity, such as "Induced Chirality" in Figure 4, "Cyclic" in Figure 5, "Host-Guest" in Figure 7, "ab initio" and "Exciton Chirality" in Figure 8, "Chiroptical Solutions" in Figure 11. Also I cannot understand the meaning to include “Classical Electromagnetism” as a separate sub-chapter and show it in Figure 8 if it is not a subject of this review.

We have revised all abbreviations and defined them the first time they appear. Figure 4 has been modified to clarify the concept “chirality induction”. To clarify the concept “cyclic” in figure 5, the figure has been modified and the text referred here to macrocycles, has been included to the manuscript. Both, Host-Guest and Sergeants and Soldiers have been modified in Figure 7. Figure 8 has been modified and the following text included in the manuscript, On the other hand, for systems presenting surface plasmon resonance, methodologies based on classical electromagnetism are often used, we refer the reader to them for a more specific reading on the topic.[64] Section 3.3 has been removed from the manuscript. Figure 11 has been improved.

The same problem with “chiroptical symmetry analysis.”

The following text has been added. For instance, for systems presenting D3 and D4 symmetry bearing three and four isolated crhomophores respectivelly, only one A2 transition and two degenerated E transitions emerge. Like in the case of exciton coupling between two independent chromophores, the prediction of the chiroptical spectrum for these systems can be performed by the following spets: i) identification of the chromophores originating the chiroptical responses, ii) using group theory, obtain the symmetry irreducible representations to determine the allowed transitions, iii) Considering the relative orientation between the chromophores, and using the Davydov´s equation, determine the energy difference between the different transitions. iv) Also with the geometric parameters, predict the electron transition and magnetic dipole moment of each transition, and consequently the rotary strength. As a case study, the chiroptical symmetry analysis of trianglimines has been recently performed.[63] The modifications are highlighted in the manuscript.

There is a large number of typos and errors in the text, which should be carefully checked.

The text has been revised and, the modifications are highlighted in the manuscript.

Round 2

Reviewer 1 Report

The authors made some changes to the previous version, improving the figures and adding comments about the results in the literature, but I still note big flaws in the text and in the explanation:

the definition of circular dichroism, one of the main subjects of the work, is completely ambigous. In lines 42-46 the authors state "the circularly polarized light-waves of opposite helicity, yet the same amplitude and phase, constitute the linearly polarized light." This is not completely correct - what phase are they referring to? A circular polarization is made of two perpendicular linear polarizations of the same amplitude, but with a phase difference of 90degrees. Then a linear polarization can be though of as a combination of two opposite circular polarizations. You need to define well terms such as helicity and phase difference. "interaction of two enantiomeric molecules with the same helical wave may be different". Definition of circular dichroism usually involves one enantiomer and both circular polarizations. Then the CD sign inverts for the opposite enantiomer. This is a big issue of the manuscript - the basic terms should be well explained. "a type of chiroptical response that can be used for several applications" (Fig. 3). The authors do not cite any literature where several applications are mentioned. They do not even mention those several applications, except  in the Figure. In the figure just sketches are shown, without any comment. One in general dedicates a whole paragraph for such statements, explaining in which way CD can be used in the applications mentioned in the sketches. What is the object in Fig. 4? Is it a molecule? System of molecules? which kind of molecules? Fig. 5 - under every sketch an example from the literature should be mentioned. Fig. 6: not clear, even from the text.

Author Response

The authors made some changes to the previous version, improving the figures and adding comments about the results in the literature, but I still note big flaws in the text and in the explanation:

We are glad to know that Reviewer 1 finds our manuscript now improved. We would like to thank her/his effort on the revision of our manuscript.

the definition of circular dichroism, one of the main subjects of the work, is completely ambiguous. In lines 42-46 the authors state "the circularly polarized light-waves of opposite helicity, yet the same amplitude and phase, constitute the linearly polarized light." This is not completely correct - what phase are they referring to? A circular polarization is made of two perpendicular linear polarizations of the same amplitude, but with a phase difference of 90degrees. Then a linear polarization can be thought of as a combination of two opposite circular polarizations.

We apologize for this inaccuracy on the definition of circularly polarized light. We have corrected the text that now reads: “In fact, circularly polarized light can be right-handed or left-handed, referred to as right circularly polarized light (R-CPL) and left CPL (L-CPL) respectively. These chiral waves can be generated by the summation of two perpendiculars linearly polarized lights with the same amplitude and frequency, yet, exhibiting p/2 phase difference. Moreover, a linearly polarized light emerges from the combination of two enantiomeric CPLs.” The modifications are highlighted in the manuscript.

You need to define well terms such as helicity and phase difference. "interaction of two enantiomeric molecules with the same helical wave may be different". Definition of circular dichroism usually involves one enantiomer and both circular polarizations. Then the CD sign inverts for the opposite enantiomer.

We apologize for this imprecision. We have corrected the text that now reads: “Chiroptical spectroscopies, which emerge from the difference in absorption and speed of R-CPL and L-CPL upon interaction with a chiral molecule are phenomena known as birefringence and circular dichroism (CD), respectively. In general, the chiroptical responses of enantiomers are opposite to each other.[5]” The modifications are highlighted in the manuscript.

This is a big issue of the manuscript - the basic terms should be well explained. "a type of chiroptical response that can be used for several applications" (Fig. 3). The authors do not cite any literature where several applications are mentioned. They do not even mention those several applications, except in the Figure. In the figure just sketches are shown, without any comment. One in general dedicates a whole paragraph for such statements, explaining in which way CD can be used in the applications mentioned in the sketches.

We apologize for this inaccuracy. We have corrected the text that now reads: “The outcome of birefringence is the rotation of the plane of polarization of a wave when traveling through a chiral system, commonly known as optical rotation. While originally these measurements were performed only at 589 nm, the D-line of sodium, resulting in the α_D values characteristic of any chiral molecule, later on, the measurement of the optical rotation at different wavelengths provided optical rotatory dispersion (ORD), a more reliable methodology for structural determination.[5,6] On the other hand, CD can be measured in the infrared region, like is the case for vibrational CD (VCD), or in the ultra violet visible region of the spectrum, with spectroscopies such as electronic CD (ECD), circularly polarized luminescence, or Raman optical activity.[7] In general, the chiroptical responses of enantiomers are opposite to each other.[8]” and “as it will be described below, (Figure 3).[9] Often, the use of more than one chiroptical response is required in order to confidentially determine the structure of a particular system.[10]” The modifications are highlighted in the manuscript.

What is the object in Fig. 4? Is it a molecule? System of molecules? which kind of molecules?

To clarify the meaning of Figure 4, the following text has been added: “Since chirality can be manifested in many different ways, in this section we group chiroptical systems originating form three different strategies: Individual, which is exhibited by helical molecules,[16] or polymers[17] or helical nanoparticles,[18]; Collective, where chirality is originating from the relative orientation of individual molecules[19] or nanoparticles;[20] and Induced Chirality, were a chiral or achiral molecule undergoes a conformational change to adopt a chiral conformation (Figure 4).[21] [21]” The modifications are highlighted in the manuscript.

Fig. 5 - under every sketch an example from the literature should be mentioned.

The text has now been implemented to indicate the specific examples that are included in the literature examples. The text now reads: “In this section we want to call the attention to systems that independently present chirality. Considering individually chiral systems one may identify them as being from molecular or nanoparticle nature (Figure 5). Very often the open molecular systems are monodisperse, like enantiopure alleno-acetylenic oligomers,[10] however, there are several examples of open polydisperse oligomers presenting chiroptical responses, for instance poly(phenylacetylene) amines.[23] The search for specific molecular recognition sites and restriction of the conformational space has resulted in = development of cyclic, like spirobifluorene macrocycles,[24] cage-like organic, such as covalent organic helical cages,[21] and organometallic systems, like alleno-acetylenic helicages.[25] On the other hand, a tremendous expansion in the last years has been realized in the development of intrinsically chiral metal nanoparticles. While for clusters, the system can be monodisperse like Au25,[26] larger nanoparticles are typically polydisperse and can be classified as 2D chiral nanoparticles where the chirality comes from the nanoparticle lying onto a surface, or 3D chiral nanoparticles. For a more specific review on chiroptical nanoparticles we refer the reader to reference [28] ” The modifications are highlighted in the manuscript.

Fig. 6: not clear, even from the text.

To clarify the meaning of Figure 6, the following text has been added: “This reality can be realized by self-assembly of independent molecules were the chirality is transfer from the individual units to the self-assembly or by template strategies were achiral units adopt relative chiral orientations by following the guidance of a chiral support (Figure 6).”

Reviewer 3 Report

In general, the manuscript has been considerably improved. However, some minor points should be properly addressed as follows.

The copyright issue; the authors should acknowledge the corresponding copyrights even if the original figures are partly reproduced. Still there are many grammar mistakes and typos in the highlighted text, which should be carefully checked either by the authors or during the editorial preparations.

Author Response

In general, the manuscript has been considerably improved. However, some minor points should be properly addressed as follows.

We are glad to know that Reviewer 3 finds our manuscript now improved. We would like to thank her/his effort on the revision of our manuscript.

The copyright issue; the authors should acknowledge the corresponding copyrights even if the original figures are partly reproduced. Still there are many grammar mistakes and typos in the highlighted text, which should be carefully checked either by the authors or during the editorial preparations.

For figures associated with infused examples, we are in the process of asking for copyright permissions. Additionally, the corresponding citations have been added to the caption of the figures. The modifications are highlighted in the manuscript.